# Investigation of Antibacterial and Antiinflammatory Activities of Proanthocyanidins from *Pelargonium sidoides* DC Root Extract

**DOI:** 10.3390/nu11112829

**Published:** 2019-11-19

**Authors:** Aiste Jekabsone, Inga Sile, Andrea Cochis, Marina Makrecka-Kuka, Goda Laucaityte, Elina Makarova, Lia Rimondini, Rasa Bernotiene, Lina Raudone, Evelina Vedlugaite, Rasa Baniene, Alina Smalinskiene, Nijole Savickiene, Maija Dambrova

**Affiliations:** 1Medical Academy, Lithuanian University of Health Sciences, Sukileliu Ave. 13, LT-50162 Kaunas, Lithuania; 2Latvian Institute of Organic Synthesis, Aizkraukles Str. 21, LV1006 Riga, Latvia; 3Riga Stradins University, Dzirciema Str. 16, LV1007, Latvia; 4Department of Health Sciences, University of Piemonte Orientale, Via Solaroli 17, 28100 Novara, Italy; 5Interdisciplinary Research Center of Autoimmune Diseases, Center for Translational Research on Autoimmune and Allergic Diseases–CAAD, C.so Trieste 15A, 28100 Novara, Italy; 6Clinic of dental and oral pathology, LSMU Hospital, Kaunas Clinics, Medical academy, Lithuanian University of Health Sciences, Eiveniu Str. 2, LT-50161 Kaunas, Lithuania

**Keywords:** periodontitis, *Pelargonium sidoides* DC root extract, proanthocyanidins, bacteriotoxicity, inflammatory cytokines, gene expression, fibroblasts, macrophages, leukocytes

## Abstract

The study explores antibacterial, antiinflammatory and cytoprotective capacity of *Pelargonium sidoides* DC root extract (PSRE) and proanthocyanidin fraction from PSRE (PACN) under conditions characteristic for periodontal disease. Following previous finding that PACN exerts stronger suppression of *Porphyromonas gingivalis* compared to the effect on commensal *Streptococcus salivarius*, the current work continues antibacterial investigation on *Staphylococcus aureus*, *Staphylococcus epidermidis*, *Aggregatibacter actinomycetemcomitans* and *Escherichia coli.* PSRE and PACN are also studied for their ability to prevent gingival fibroblast cell death in the presence of bacteria or bacterial lipopolysaccharide (LPS), to block LPS- or LPS + IFNγ-induced release of inflammatory mediators, gene expression and surface antigen presentation. Both PSRE and PACN were more efficient in suppressing *Staphylococcus* and *Aggregatibacter* compared to *Escherichia*, prevented *A. actinomycetemcomitans*- and LPS-induced death of fibroblasts, decreased LPS-induced release of interleukin-8 and prostaglandin E2 from fibroblasts and IL-6 from leukocytes, blocked expression of IL-1β, iNOS, and surface presentation of CD80 and CD86 in LPS + IFNγ-treated macrophages, and IL-1β and COX-2 expression in LPS-treated leukocytes. None of the investigated substances affected either the level of secretion or expression of TNFα. In conclusion, PSRE, and especially PACN, possess strong antibacterial, antiinflammatory and gingival tissue protecting properties under periodontitis-mimicking conditions and are suggestable candidates for treatment of the disease.

## 1. Introduction

Periodontitis is an infectious inflammatory disease resulting in periodontal pocket formation, progressive bone reduction and teeth loss in many industrialized countries [1,2]. Common treatment strategies include systemic use of antibiotics and local synthetic antiseptic substances, both leading to undesirable side effects and increased resistance of bacteria [3]. In consequence, prolonged and/or repeatable treatment is risky, inefficient and fails to stop disease remission and further progression. In fact, as a response to the extensive use of drugs, bacteria have developed a new mechanism to skip and counteract antibiotics activity: resistant polysaccharide envelope, more efficient efflux pumps, intracellular modifications and genetic mutations are some of the pathways exploited by bacteria to withstand drugs effect [4]. However, it is important to consider that not all body-resident bacteria are pathogens: commensal strain present in the microbiota play a pivotal role in preserving homeostasis in the skin and mucosal physiological systems of the human body [5,6]. The use of very strong chemicals such as chlorhexidine [7] can be exploited only for short periods to prevent severe side effects that can occur after prolonged exposure [8]. It follows that an ideal new antibacterial compound should be able to affect bacteria metabolism by a different mechanism than those exploited by antibiotics but at the same time would be harmless to the healthy cells and commensal bacteria. In this light, multicomponent plant-derived antibacterial substances like proanthocyanidins (PACN) make a promising alternative and adjunctive therapy candidates for periodontitis treatment because of a lower risk of resistance development and side effects [9]. 

PACN are condensed tannins constructed form flavan-3-ol units [10]. The compounds possess a range of biological activities including anti-inflammatory and antibacterial [11]. The capacity of PACN to suppress inflammation is related to both strong antioxidant and metalloproteinase (MMP) inhibiting properties [12,13], whereas antibacterial efficiency is achieved due to prevention of bacterial adhesion and biofilm formation [14]. The chemical nature of PACN in crude extracts varies depending on plant species used. *Pelargonium sidoides* DC, a medicinal plant native to South Africa, is one of the most PACN-enriched plants. Medicinal raw materials—roots of the plant—are used in the treatment of infectious and inflammatory disorders, and *P. sidoides* root extracts (PSREs) possess the same properties with enhanced efficiency [15,16,17,18]. PSREs mediate their pharmacological effects via two classes of compounds, namely oxygenated coumarins and prodelphinidins that belong to the PACN group [18]. The common properties of these compounds isolated from various sources suggest the significant part of the activities of PSREs might be assigned to PACN. Indeed, we have recently shown that namely prodelphinidin fraction from PSRE more efficiently suppress periodontal pathogens *Porphyromonas gingivalis* compared to PSRE itself [19]. Moreover, the activity appeared to be strain selective: reducing the viability of the pathogens while preserving the metabolic activity of the beneficial oral commensal *Streptococcus salivarius*.

Based on these promising results, in the present study we decided to extend the examination of antibacterial efficiency of PACN to other broad-range pathogens and commensals: two commercial drug-resistant *Staphylococcus aureus* and *Aggregatibacter actinomycetemcomitans* strains, a clinical isolate pathogen *Staphylococcus epidermidis* strain and a commensal *Escherichia coli* strain. Next, after verifying extract cytocompatibility towards gingival fibroblasts, a “race for the surface” model of bacteria-cells co-culture [20] was carried out to verify the extract ability to reduce bacteria proliferation while preserving cells viability in the same microenvironment where cells and bacteria compete for the same surface. Finally, we have made an extensive investigation on PACN activity in bacterial lipopolysaccharide (LPS)-mediated inflammation, including measurement of secretion of inflammatory cytokines and other mediators, inflammatory gene expression and viability of gingival fibroblasts, macrophages and blood leukocytes.

## 2. Materials and Methods

### 2.1. Pelargonium sidoides Root Extract and Proanthocyanidin Fraction

The *P. sidoides* root extract (PSRE) was purchased from Frutarom Switzerland Ltd. Rutiwisstrasse 7 CH-8820 Wadenswil (batch no. 0410100). Proanthocyanidins (PACN) from PSRE were purified as described by Hellström and co-authors [21] with some modifications [19]. Briefly, 4 g of PSRE was dissolved in 200 mL of 50% methanol, the solution was centrifuged at 2000× *g* for 20 min and filtered through 0.45 µm nylon filters. The solution was purified by gel adsorption over Sephadex LH-20. The proanthocyanidins were released from the gel with 70% aqueous acetone (500 mL) and concentrated under vacuum at 35 °C. The aqueous aliquot was freeze-dried. The freeze dried PACN preparation yielded in 1.37 ± 0.07 g and comprised about 34.25% of the loaded PSRE.

### 2.2. Bacterial Strains and Growth Conditions

Commercially available strains *Staphylococcus aureus* (*S. aureus*, pathogen, ATCC 43300), *Aggregatibacter actinomycetemcomitans* (*A. actinomycetemcomitans*, pathogen, ATCC 33384) and *Escherichia coli* (*E. coli*, non-pathogen, ATCC BAA-1427) were purchased from the American Type Culture Collection (ATCC, MA, USA) and cultivated following the manufacturer’s instructions. A clinical isolate of *Staphylococcus epidermidis* (*S. epidermidis*, pathogen) was collected at the Clinical Microbiology Unit at the Novara Maggiore Hospital (Novara, Italy). The clinical isolate was obtained after patient’s informed consent in full accordance with the Declaration of Helsinki. Clinical strain was cultivated in Luria-Bertani medium (LB, Sigma-Aldrich, Milan, Italy) at 37 °C. For experiments, a single colony from each strain was collected and inoculated in 9 mL of LB broth at 37 °C overnight (18 h). After incubation, a new fresh LB tube diluted 1:10 was prepared and incubated at 37 °C for 3 h to achieve the logarithmic growth phase. Finally, broth cultures were diluted in LB broth until the optical density was 0.001 at 600 nm, corresponding to a final concentration of 1 × 10^5^ cells/mL.

### 2.3. Antibacterial Efficiency Evaluation

To test antibacterial activity, PSRE and PACN were used at the following concentrations: 10, 30, 50, 70, 80, 90 and 100 μg/mL. PSRE and PACN powders were mixed and diluted directly into LB medium containing the desired bacteria concentration (described in 2.2. chapter); 1 mL of the obtained mix solutions (LB containing bacteria + PSRE/PACN) was seeded in the wells of a 24 multiwell plate (SPL, BioSigma, Milan, Italy) and incubated 24 h at 37 °C. Bacteria cultivated into pure LB medium were considered as control. After incubation, bacteria viability was evaluated by means of the colorimetric metabolic assay Alamar blue (alamarBlue^®^, Thermo-Fisher, Waltham, MA, USA) following the manufacturer’s instructions. Briefly, the ready-to-use solution was added to each well in a 1:10 ratio and incubated for 4 h in the dark at 37 °C; then, fluorescence was recorded at 590 nm using a spectrophotometer (Spark, Tecan, Basel, Switzerland).

### 2.4. Co-Cultures of Human Gingival Fibroblasts and Bacteria

To verify PSRE and PACN ability to preserve cells metabolism in the presence of infection, a race for the surface cells-bacteria co-culture experiment was set up. *S. aureus* and *A. actinomycetemcomitans* were selected and used with cells to simulate the oral and mucosal environments, respectively. Human primary gingival fibroblasts (HGF, ATCC PCS-201-018) were used as a cellular model to be tested with bacteria. Cells were seeded at a defined number (1.5 × 10^4^ cells/well) in the wells of a 24 multiwell plate and allowed to adhere overnight. Afterwards, the medium was removed and replaced by 1 mL of solution composed by an antibiotics-free medium (minimal essential medium Eagle alpha-modification, from Sigma) supplemented with 10% fetal bovine serum (FBS, Sigma) and 100 μg/mL of either PSRE or PACN and 1 × 10^5^ bacteria. The plate was incubated 24 h at 37 °C to allow cells-bacteria direct contact. Then, cells and bacteria were detached from plate wells by a collagenase (1 mg/mL) trypsin-EDTA (0.25%) solution and collected. The number of viable cells was determined by trypan blue and Burker chamber count. Cells cultivated with fresh medium without bacteria were considered as a control.

### 2.5. Rat Gingival Fibroblast Cell Culture and Treatments

All experimental procedures were performed according to the Law of the Republic of Lithuanian Animal Welfare and Protection (License of the State Food and Veterinary Service for working with laboratory animals No. G2-80). The mice were maintained and handled at Lithuanian University of Health Sciences animal house in agreement with the ARRIVE guidelines. Primary gingival fibroblasts were isolated from gingiva of P5-7 rat pups. After isolation, the cells were grown in 75 cm^2^ flasks in DMEM with high glucose and Glutamax (Thermo Fisher Scientific, Waltham, MA, USA), 10% FBS and Pen/Strep. At 70%–90% confluence, the cells were detached by 0.025% Trypsin/EGTA and plated in 96 well plates at a density of 2 × 10^5^ cells/well. The treatments were made 24 h after plating. All treatments were made simultaneously, without pre-incubations, and lasted 24 h. LPS was used at concentration of 2 mg/mL (1 mg/mL LPS did not induce significant increase in cell death), filtered PSRE and PACN solutions at 50 and 100 μg/mL (both preparations induced toxicity starting from 200 μg/mL).

### 2.6. Bone Marrow-Derived Macrophages

For bone marrow-derived macrophages (BMDM) isolation male C57BL6/J inbred mice (18–20 weeks old, Envigo, Netherlands) were used. The experimental procedures were carried out in accordance with the guidelines of the European Community (2010/63/EU), local laws and policies and were approved by the Latvian Animal Protection Ethical Committee, Food and Veterinary Service, Riga, Latvia. Mice were euthanized by decapitation, and bone marrow cells were extracted from femur bones and differentiated for 7 days in RPMI-1640 with Glutamax (Gibco,) supplemented with 10% FBS, 1% antibiotics and 10 ng/mL M-CSF (monocyte-colony stimulating factor, PeproTech, London, UK). Then cells were detached by 0.5% trypsin (Sigma Aldrich), and plated in a 12-well plate (11 × 10^5^ cells/mL) in DMEM-high glucose medium supplemented with 10% FBS, 1% antibiotics. After 1h incubation in 37 °C incubator, cells were stimulated with PSRE and PACN at 100 µg/mL and LPS 10 ng/mL with murine IFNγ (interferon gamma, PeproTech) 100 U/mL for proinflammatory gene expression and macrophage polarization to M1 (pro-inflammatory) phenotype for 2 h and 24 h, respectively.

### 2.7. Human Peripheral Blood Mononuclear Cells

Human peripheral blood mononuclear cells (PBMCs) were purchased from ATCC (ATCC^®^ PCS-800-011™, Manassas, VA, USA). The cells were cultured at 3.3 × 10^6^ cells/mL (12 well plate) in RPMI medium supplemented with 10% FBS, 1% antibiotics. After 1 h incubation in 37 °C incubator, cells were stimulated with 1 µg/mL LPS (lipopolysaccharide, Sigma-Aldrich) in the presence of 100 µg/mL PSRE or PACN, for 6 h.

### 2.8. Analysis of Cell Viability by Lactate Dehydrogenase Release, Alamarblue and MTT Assay

PBMCs’ viability was assessed by measuring lactate dehydrogenase (LDH) release in cell culture media. LDH activity was measured using a method based on the reduction of a tetrazolium salt (yellow) to formazan (red) [22]. The absorbance of kinetic parameters was determined spectrophotometrically at 503 nm on Hidex Sense microplate reader. The reaction mixture contained 30 mM lactate, 150 µM NAD+ (nicotinamide adenine dinucleotide), 0.4 mM 2-(4-iodophenyl)3-(4-nitrophenyl)-5-phenyltetrazolium chloride (INT) and 3.25 mM *N*-methylphenazonium methyl sulphate (PMS) in 100 mM Tris buffer solution (pH 8.0). In addition, PBMCs viability after 24 h incubation with different concentrations of PSRE and PACN was determined with alamarBlue^®^, (Bio-Rad Laboratories, Hercules, CA, USA) following the manufacturer’s instructions. Briefly, the ready-to-use solution was added to each well in a 1:10 ratio and incubated for 2 h in the dark at 37 °C; then, fluorescence (Ex 544 nm/Em 590 nm) and optical density (570 and 600 nm) using Hidex Sense microplate reader. 

BMDM viability after 24 incubation with different concentrations of PSRE and PACN was determined using MTT assay. After incubation, BMDM were incubated with MTT (TCI Europe) solution (1 mg/mL) for 1 h, the formazan crystals formed during incubation were dissolved in isopropanol, and optical density at 570 nm corresponding to the amount of viable cells was measured in a Hidex Sense microplate reader.

### 2.9. Necrosis Evaluation by Double Nuclear Staining

The level of necrotic cell death was assessed by double nuclear fluorescent staining with Hoechst33342 (10 mg/mL) and propidium iodide (PI, 5 mg/mL), 5 min at 37 °C. PI-positive nuclei indicating lost nuclear membrane integrity were considered necrotic. Cells were visualized under fluorescent microscope OLYMPUS IX71S1F-3 (Olympus Corporation, Tokyo, Japan), counted in fluorescent micrographs and expressed as the percentage of total cell number per image. The data are presented as averages ± standard deviation. LD50 was calculated by SigmaPlot v.13 (Systat Software Inc, London, UK) using the equation proposed by a dynamic curve fitting tool.

### 2.10. Apoptosis Evaluation by Annexin V staining

The viable cells were analyzed on a BD FACS Melody™ (BD Biosciences, San Jose, CA, USA) flow cytometer using Annexin V-allophycocyanin (BioLegend, San Diego, CA, USA) staining. For analysis of BMDMs apoptosis, treated cells were stained with fluorescent annexin V antibody and, afterwards, the proportion of apoptotic (Annexin V positive) cells was evaluated.

### 2.11. Caspase Activity Assessment

Caspase-3 activity in cell lysates after treatments was measured by means of a Caspase 3 Assay Kit (Sigma-Aldrich) according to the manufacturer’s protocol, by assessing Ac-DEVD-7-amido-4-methylcoumarin cleavage and subsequent increase in 7-amido-4-methylcoumarin (AMC) fluorescence in a fluorometric plate reader Ascent Fluoroskan (Thermo Fisher Scientific, Waltham, MA, USA; λ_ex_ = 360 nm, λ_em_ = 460 nm). The data are presented as averages ± standard deviation of AMC concentration increase rate normalized for mg of cellular protein, averages ± standard deviation.

Caspase-8 in cell lysates after treatments was measured by means of a colorimetric Caspase 8 Assay Kit (Sigma-Aldrich) according to the manufacturer’s protocol, by assessing Ac-IETD-*p*-Nitroaniline cleavage and increase in *p*-Nitroaniline (pNA) concentration in a spectrophotometric plate reader Multiskan Go 1510 (Thermo Fisher Scientific, Waltham, MA, USA) reading absorbance at λ_ex_ = 405 nm. The data are presented as averages ± standard deviation of pNA concentration increase rate normalized for mg of cellular protein, averages ± standard deviation.

### 2.12. Detection of Secreted Inflammatory Mediators

Medium collected after treatments was assayed for cytokines tumor necrosis factor-α (TNF-α), interleukin-6 (IL-6), interleukin-8 (IL-8) and prostaglandin E2 (PGE2) production using TNF-α mouse (Millipore), IL-6 human (Sabbiotech), IL-8 rat (Abbexa) and PGE2 rat (Abbexa) kits following the manufacturer’s protocols.

### 2.13. Bone Marrow-Derived Macrophage Polarisation to M1 Phenotype and Analysis by Flow Cytometry

BMDMs were incubated with PSRE and PACN (100 μg/mL each) and LPS + IFNγ (10 ng/mL/100 U/mL) for 24 h. The cells were washed twice with HBSS and harvested by trypsin (0.5%), then DMEM-high glucose medium with 10% FBS was added and cell suspension was centrifuged at 300× *g* for 5 min. Then cells were incubated with specific conjugated antibody mixtures (in concentration 1:100 in cell wash buffer) for 30 min on ice in the dark. The mixture contained following monoclonal antibodies purchased from BioLegend (San Diego, CA, USA): FITC-conjugated anti-mouse F4/80, phycoerythrin (PE)-conjugated anti-mouse CD86 and biotin-conjugated anti-mouse CD80. Then cells were washed and stained with streptavidin-APC-Cy7. After 30 min, cells were washed and stained for evaluation for apoptosis (see method above). After staining samples were analyzed by flow cytometry (BD FACS Melody™, BD Biosciences, San Jose, CA, USA).

### 2.14. mRNA Isolation and Quantitative RT-PCR Analysis

Total RNA from cells was isolated using Ambion PureLink RNA Mini Kit (catalogue No. 12183025) according to the manufacturer’s protocol. The first-strand cDNA synthesis was carried out using a High Capacity cDNA Reverse Transcription Kit (Applied Biosystems^TM^, Foster City, CA, USA) following the manufacturer’s instructions. The quantitative RT-PCR analysis of gene expression was performed by mixing SYBR Green Master Mix (Applied Biosystems^TM^), synthesized cDNA, forward and reverse primers specific for interleukin-1β (IL-1β), interleukin 10 (IL-10), inducible nitric oxide synthase (iNOS), TNF-α, cyclooxygenase 2 (COX-2) and running the reactions on a Mic Real-Time PCR instrument. The relative expression levels for each gene were calculated with the ∆∆Ct method and normalized to the expression of glucose-6-phosphate isomerase gene.

### 2.15. Statistical Analysis

The quantitative results are presented as mean ± standard deviation (SD) of 3–7 replicates. The data were processed using Microsoft Office Excel 2010 (Microsoft, Redmond, WA, USA) and SPSS 20 (IBM, Armonk, NY, USA) software. For antibacterial activity testing and bacterial-fibroblast co-culture experiments, the statistical data analysis was performed by applying the ANOVA with a Tukey HSD post hoc test. For experiments on rat gingival fibroblasts, ANOVA with Dunn’s test was used and data analyzed by SigmaPlot v.13 (Systat Software Inc., London, UK). In all cases, differences were considered statistically significant when *p* < 0.05.

## 3. Results

### 3.1. Antibacterial Activity of Proanthocyanidins from Pelargonium sidoides Root Extract

For antibacterial efficiency evaluation, bacterial strains *S. aureus*, *S. epidermidis*, *A. actinomycetemcomitans* and *E. coli* were subjected to PSRE and PACN treatments were used at the concentration ranging from 10 to 100 μg/mL. Antibacterial activity results are summarized in Figure 1.

Both PSRE and PACN significantly reduced bacterial metabolic activity in comparison to the untreated control, starting from concentrations of 50–70 μg/mL. However, some differences were noticed between the tested strains. In the case of *S. aureus*, PSRE was more effective than PACN (Figure 1a). Fifty μg/mL of PSRE was enough to significantly decreased metabolic activity of this strain, whereas PACN had a similar result at 80 μg/mL (*p* < 0.05 vs. control, indicated by the *). For *A. actinomycetemcomitans*, the results were opposite (Figure 1d); PACN significantly decreased metabolic activity at 50 μg/mL, but it required 80 μg/mL PSRE to achieve this level of activity. The metabolism of the clinical isolate *S. epidermidis* (Figure 1b) was significantly decreased by both PSRE and PACN applied at a 70 μg/mL concentration.

Interestingly, the non-pathogen *E. coli* demonstrated the highest resistance to the treatment (Figure 1c) when the 0–80 μg/mL window was considered. After the 80 μg/mL PSRE treatment, metabolic activity of *E. coli* decreased by 26% compared with untreated control, whereas metabolic activity of *S. aureus, S. epidermidis* and *A. actinomycetemcomitans* was reduced by 75%, 97% and 57%, accordingly. In the presence of 80 μg/mL PACN, the loss of metabolic activity of these four strains listed in the same order was 47%, 74%, 96% and 99%. Moreover, even after addition of 100 μg/mL PACN to the growth medium, *E. coli* still preserved nearly half of the control activity level, but in the case of *A. actinomycetemcomitans*, the twice less amount of this preparation almost completely blocked bacterial growth. Summarizing, the results of the antibacterial evaluation indicate stronger toxicity of PSRE and PACN to *S. aureus*, *S. epidermidis* and *A. actinomycetemcomitans* than to non-pathogenic *E. coli* strain when the extracts were used within the 80 μg/mL range. Conversely, at higher 90 and 100 μg/mL concentrations a similar broad range effect was observed for all the tested strains as all results of both PSRE and PACN were significantly different in comparison with untreated controls (*p* < 0.05, indicated by the *).

### 3.2. The Effect of Proanthocyanidins from Pelargonium sidoides Root Extract on Gingival Fibroblast Viability under Conditions of Bacterial Infection

#### 3.2.1. *Pelargonium sidoides* Root Extract and Proanthocyanidins Preserve Gingival Fibroblasts in the Presence of Bacteria

Next in the study, we investigated whether antibacterial properties of PSRE and PACN are efficient in protecting human gingival fibroblasts during bacterial infection. For this purpose, two microenvironments were simulated: oral, by growing together gingival fibroblast cells and bacteria *A. actinomycetemcomitans*, and mucosal, by co-culturing gingival fibroblasts and *S. aureus*. Results obtained by the co-culture “race for the surface” simulation are reported in Figure 2.

Both the presence of PSRE and PACN at 100 μg/mL were effective in protecting cells from bacterial infection; the number of viable cells counted after 24 h of direct contact with bacteria was comparable with the control consisting of cells cultivated in fresh medium without bacterial presence. Conversely, when bacteria were cultivated in the same wells of cells but in the absence of PSRE or PACN (no treatment), they were able to fully colonize the well. In this scenario, no viable cells were detected for both applied models. The results indicate that the presence of either of the substances was effective in preserving gingival fibroblast cell viability by lowering bacteria proliferation.

#### 3.2.2. *Pelargonium sidoides* Root Extract and Proanthocyanidins Protect Gingival Fibroblasts from Necrotic Cell Death Induced by Bacterial Lipopolysaccharide

The protective effect of PSRE and PACN under conditions of bacterial infection might be mediated not only by suppression of bacterial growth, but also by increasing the resistance of the cells to bacterial metabolites. Therefore, we examined how PSRE and PACN affect the viability of primary murine gingival fibroblasts in the presence of bacterial LPS. To select the concentrations of PSRE and PACN that has no harmful effect for the cells, the concentration-dependent toxicity was defined by double nuclear staining with propidium iodide and Hoechst33342.

The level of necrotic cells with propidium iodide-positive nuclei significantly increased after 24 h treatment with 200 μg/mL and higher concentration of both PSRE and PACN (Appendix A). PSRE demonstrated significantly higher toxicity compared to PACN in the concentration range between 200 and 350 μg/mL, and this was reflected by established LD50 for the preparations: 209 μg/mL for PSRE and 288 μg/mL for PACN. In the concentration range from 400 to 550 μg/mL, there were no further difference between the effect of both preparations and nearly all cells in the culture were found necrotic. Based on these data, there were two concentrations of PSRE and PACN selected for antiinflammatory effect study: 50 and 100 μg/mL. Both solutions were not toxic to gingival fibroblasts when applied for 24 h at these concentrations.

After treatment with 2 μg/mL LPS, the amount of necrotic cells with propidium iodide-positive nuclei in fibroblast cell culture increased from 2.7% ± 2.2% in control to 31.7% ± 11.9%, reflecting a statistically significant loss in cell viability (Figure 3). However, the treatments with 50 and 100 μg/mL PSRE and PACN prevented cell viability loss induced by LPS. The data indicate that both PSRE and PACN could protect gingival fibroblasts from bacterial LPS-induced necrosis.

#### 3.2.3. *Pelargonium sidoides* Root Extract and Proanthocyanidin Fraction Prevent Caspase Activity Induced by Bacterial Lipopolysaccharide

Next to necrosis, bacterial toxicity might initiate apoptosis that also contributes to the loss of gingival tissue. Cysteine aspartic proteases (caspases) are key mediator enzymes in apoptosis [23]. There are two main groups of caspases according to their place in the apoptotic event cascade: initiator caspases acting as apoptotic triggers and effector caspases that are amplified by triggers and execute proteolysis leading to the characteristic biochemical and morphological changes in the apoptotic cell. To determine apoptosis-preventing capacity of PSRE and PACN, we assessed the level of active initiator caspase-8 and effector caspase-3 in LPS-treated fibroblast cells.

A 24 h LPS treatment induced elevation in caspase-8 substrate cleavage activity; it increased from 0.007 ± 0.001 nmol/min/mg in control samples to 0.047 ± 0.013 nmol/min/mg (Figure 4a). PSRE at a 50 μg/mL concentration did not significantly affect LPS-induced caspase-8 activity. However, treatments with 100 μg/mL PSRE, 50 μg/mL PACN and 100 μg/mL PACN significantly prevented caspase-8 activation by LPS by 2.2, 2.0 and 5.9 times, respectively. Treatments with 100 μg/mL PSRE and PACN without LPS had no significant effect on the activity of the caspase.

After treatment with LPS, effector caspase-3 substrate cleavage rate increased from 0.012 ± 0.01 to 0.189 ± 0.06 nmol/min/mg (Figure 4b). Both PSRE and PACN at all tested concentrations significantly reduced the effect of LPS on caspase-3 activity. When LPS was applied together with 50 μg/mL PSRE, caspase-3 activity was nearly two times lower than in LPS only-treated samples. 100 μg/mL PSRE decreased the effect of LPS on caspase-3 activity 4.7 times. Fifty and 100 μg/mL PACN lowered LPS-induced caspase-3 activity 3 and 8.8 times, respectively. There was no significant effect on caspase-3 substrate cleavage rate observed after treatment with 100 μg/mL PSRE or PACN alone, without LPS.

Overall, caspase activity evaluation indicates that both PSRE and PACN could suppress apoptotic protease activity evoked by bacterial LPS. Both preparations have a stronger effect in decreasing the executing caspase-3 activity compared with the effect on trigger caspase-8. However, PACN was more efficient in suppressing caspase-8 activity than PSRE, thus, PACN is a more powerful suppressor of apoptosis at the early stages. None of the substances was toxic to gingival fibroblasts when applied for 24 h at these concentrations.

### 3.3. The Effect of Proanthocyanidins from Pelargonium sidoides Root Extract on Inflammatory Responses to Bacterial Lipopolysaccharide

#### 3.3.1. The effect of *Pelargonium sidoides* Root Extract and Proanthocyanidin Fraction on Lipopolysaccharide-Induced Secretion of Inflammatory Mediators

Infection-induced inflammation is the main responsible for gingival and dental tissue loss in the pathogenesis of periodontal disease. Thus, for successful treatment of the disease it is important to defeat both infection and inflammation. Next in the study, we examined antiinflammatory properties of PSRE and PACN on LPS-induced release of inflammatory mediators from gingival fibroblasts and blood leukocytes.

Increased IL-8 production by gingival fibroblasts is responsible for attraction of neutrophils to inflamed regions and rapid tissue loss during periodontitis [24,25]. Evaluation of IL-8 amounts secreted in the medium by cultured gingival fibroblasts revealed that after 24 h with LPS the amount of the cytokine increased from nearly zero to 1022 ± 75 ng/mL (Figure 5a). In the presence of 50 μg/mL PSRE, the level of IL-8 after same LPS stimulation was only 344 ± 49 ng/mL, i.e., three times lower than without the extract. Increasing PSRE concentration to 100 μg/mL further suppressed IL-8 release to 201 ± 33 ng/mL (five times less than after LPS-only treatment). A similar suppression level was achieved by 50 μg/mL PACN, and in the presence of 100 μg/mL PACN, the level of IL-8 after LPS stimulation had further dropped to 32 ± 12 ng/mL, making the level 16 times lower than after treatment with LPS alone.

The level of PGE2, a mediator of cyclooxygenase-2 inflammatory pathway, was found to be dramatically increased in the cell culture medium after LPS treatment (Figure 5b). In control samples, the average amount of PGE2 was 5 ± 2 ng/mL, but after 24 h with LPS, it had climbed up to 2372 ± 194 ng/mL. In the presence of both 50 and 100 μg/mL PSRE, the levels of PGE2 after LPS treatment were 637 ± 58 ng/mL and 613 ± 133 ng/mL, respectively, i.e., nearly four times lower compared with the level without the extract. Treatments with 50 and 100 μg/mL PACN were even more efficient, further decreasing PGE2 level in the medium to 174 ± 41 ng/mL and 72 ± 16 ng/mL, or 14 and 33 times compared with LPS only treatment, respectively. Thus, the effects of PACN on LPS-stimulated PGE2 release were significantly stronger than those of PSRE.

Bacterial invasion also cause an infiltration of leukocytes that mediate inflammation and disturb osteoblast-osteoclast balance via release of IL-6 [23]. Thus, we investigated how PSRE and PACN affect the release of IL-6 from PBMCs. One hundred μg/mL of PSRE and PACN significantly decreased LPS-induced secretion of IL-6 from PBMCs to 67% and 18% of the level caused by LPS stimulation, respectively (Figure 5c). Note that neither PSRE nor PACN were toxic to the cells at the concentrations applied as revealed by metabolic viability analysis (Appendix A).

The data about inflammatory mediator secretion indicate that both PSRE and PACN efficiently suppress LPS-induced IL-8 and PGE2 release from gingival fibroblasts and IL-6 release from mononuclear leukocytes. PACN had slightly stronger IL-8 and IL-6 release suppressing activity, and significantly stronger PGE2 release suppressing activity than PSRE.

#### 3.3.2. The Effect of *Pelargonium sidoides* Root Extract and Proanthocyanidin Fraction on Lipopolysaccharide-Induced Expression of Inflammation-Related Genes

The release of inflammatory factors is the first step of innate immune response to pathogens. The next step leading to prolonged and enhanced inflammatory reaction is induction of inflammatory genes to produce new mediators. If uncontrolled at this stage, acute inflammation may become chronic and contribute to tissue loss in periodontitis. Activated macrophages and leukocytes are the main source of interleukin-1β (IL-1β) and TNF-α, the acute phase pyrogenic cytokines involved in most of the processes that maintain inflammation [26]. Bacterial infection also triggers inducible NO synthases (iNOS) to produce reactive nitrogen species stress on pathogens [27]. Therefore, we have examined the capacity of PSRE and PACN to modulate expression of inflammatory genes IL-1β, TNF-α and iNOS in primary murine bone marrow-derived macrophages and human mononuclear leukocytes under treatment of LPS. Stimulation of macrophages in the presence of interferon-γ (IFN-γ) acting as enhancer of LPS-induced gene expression [28] induced an increase in transcription of all the three genes investigated (Figure 6a–c).

Both preparations at a dose of 100 µg/mL significantly suppressed the mRNA transcription of IL-1β and iNOS (Figure 6a,b). The level of the IL-1β mRNA decreased by 78% of the initial level with LPS after treatment with PSRE, and by 89%—after treatment with PACN. For iNOS, the decrease in mRNA level after PSRE and PACN treatment was 53% and 64%, respectively. However, the incubation with both substances did not affect LPS plus IFN-γ-induced TNF-α gene expression (Figure 6c). 6 h treatment with LPS caused significant increase in cyclooxigenase-2 (COX-2), TNF-α and IL-1β gene transcription in human PBMCs (Figure 6d–f). PSRE and PACN at a concentration of 100 µg/mL significantly suppressed mRNA transcription of COX-2 and IL-1β. When LPS was together with PSRE, COX-2 and IL-1β mRNA levels dropped by 50%, 73% and 56%, respectively. For PACN, mRNA synthesis for these cytokines was suppressed by 63%, 89% and 76%. Similarly to BMDMs case, neither PSRE, nor PACN significantly affected TNF-α gene expression. Both PSRE and PACN at a concentration of 100 μg/mL were not toxic for PBMCs and BMDMs as revealed by metabolic activity and apoptosis evaluation (Appendix A).

The gene expression analysis indicate that both preparations acted as inflammatory signal suppressors preventing expression of pro-inflammatory cytokine IL-1β and prostaglandin producing enzyme COX-2 genes, as well as decreasing synthesis of iNOS and protecting tissues from the damage of reactive nitrogen species. However, neither PSRE nor PACN significantly influenced TNF-α gene expression.

#### 3.3.3. The Effect of *Pelargonium sidoides* Root Extract and Proanthocyanidin Fraction on Lipopolysaccharide-Induced Macrophage Conversion to M1 Phenotype

Activated macrophages can be polarized into a proinflammatory M1 phenotype and alternative anti-inflammatory M2 phenotype [29]. Increase in the M1/M2 macrophage ratio can lead from an antibacterial defense to the development of periodontitis and positively correlates with the severity of the disease [30]. Next in the study, we tested how PSRE and PACN affect LPS and IFN-γ-stimulated macrophage polarization to proinflammatory M1 phenotype characterized by the presentation of surface markers CD80 and CD86 [31,32].

Flow cytometry analysis revealed that in response to LPS and IFN-γ, the amount of M1-polarised macrophages increased 9.3 times compared to the untreated control (Figure 7). Both PSRE and PACN at a concentration of 100 µg/mL were effective in reducing the level of CD80 and CD86-positive cells. The population of cells with the exposed markers after treatment with PSRE was by 58% lower, and after treatment with PACN by 71% lower than after LPS and IFN-γ stimulation without the treatments. The results indicate that both substances were potent in preventing macrophage conversion to proinflammatory M1 phenotype under exposure to LPS and IFN-γ treatment.

## 4. Discussion

Increasing antibiotic resistance makes the search for alternative antimicrobial compounds of a crucial importance for global health [33]. Failure to defeat fast adapting pathogens without significant damage to host tissues is a key challenge in management of chronic infectious-inflammatory disease including periodontitis [34]. Progressive bacteria-driven inflammatory response causes continuous damage on periodontal cells making them more sensitive to harmful effects of antibiotics and antimicrobial chemicals [3,35]. The damage is further exacerbated by the treatment-caused loss of beneficial commensal bacteria [36]. This suggests reconsidering the possibilities of alternative treatment strategies including use of specific pathogen-targeting bacterial strains [37] and plant-derived antibacterials, because such strategies are characterized by lower or no side effects and resistance development risk, as well as complex antiinflammatory and tissue renewal stimulating properties. This study explored antibacterial and antiinflammatory properties of PSRE that is known as potent infection-defeating preparation and PSRE-derived PACN possessing stronger antioxidant and antibacterial properties compared to PSRE [17,19].

Both substances were effective in reducing metabolic activity of the selected strains suggesting a broad range of antibacterial properties. This is in line with previous evidence about various extracts prepared from *P. sidoides* roots. A commercial aqueous-ethanolic extract from *P. sidoides* EPs^®^ 7630 (Umckaloabo^®^) is reported to inhibit growth of *Streptococcus pyogenes*, *Proteus mirabilis, Staphylococcus aureus*, *Escherichia coli*, *Streptococcus pneumoniae*, *Haemophilus influenza*, *Staphylococcus epidermidis* and some other gram-negative and gram-positive bacterial strains (summarized in [38]). Aqueous-acetone PSRE was efficient in decreasing growth of antibiotic-resistant *S. aureus* strains [39]. The present study for the first time demonstrated the growth-suppressing efficiency of PSRE and PACN on *Aggregatibacter actinomycetemcomitans*, one of the most important gram-negative anaerobic periodontal pathogens [40]. Similarly as in the previously demonstrated case of *P. gingivalis* [19], PACN demonstrated significantly higher toxicity on *A. actinomycetemcomitans*, compared to the effect of PSRE. Fifty μg/mL PACN reduced metabolic activity of *A. actinomycetemcomitans* nearly 10 times more if compared to the untreated control value (Figure 1d). The same concentration had no significant toxicity on other investigated strains. The minimal amount of PACN causing a significant effect on metabolic activity of *E. coli* and *S. aureus* was 80 μg/mL, and for *S. epidermidis* the significant toxicity started from 70 μg/mL PACN. The results indicate that there might be a specific interaction of proanthocyanidins from PSRE with the main pathogenic strains (*P. gingivalis* and *A. actinomycetemcomitans*) responsible for the development of periodontitis. Strain-specific activity of proanthocyanidins was already noticed by other authors. Lacombe and Wu have reviewed the selective pathogen-suppressing and beneficial strain-promoting activity of proanthocyanidins derived from various berries [41]. However, despite many publications reporting a selective activity of natural extracts towards pathogen and non-pathogen strains, it is still not completely clear how this selection occurs [42]. It was shown that cranberry-derived proanthocyanidins are able to interfere with a *N*-acylhomoserine lactone-mediated quorum sensing of *Pseudomonas aeruginosa* [43]. Moreover, proanthocyanidins have also been shown to compromise adhesion to host cells by mimicking cell surface signaling [44]. Some authors have proposed the hypothesis that proanthocyanidins might increase bacterial membrane permeability and cause indirect metabolism decrease due to ATP and other intracellular metabolite loss [42,45]. A recent study shows that proanthocyanidins can potentiate antibiotics by acting via bacterial multidrug efflux pumps [46]. Thus, the disturbance in transmembrane transport indeed might be the cause of bacteriotoxicity. However, more studies definitely are required to clarify the mechanism of action of proanthocyanidins against pathogenic bacterial strains.

On the other hand, PSRE was more efficient than PACN in suppressing both of *Staphylococcus* strains that were investigated in this study suggesting that other than proanthocyanidin fraction compounds were acting against these bacteria. Most likely, the distinct antibacterial activity of PSRE can be ascribed to other phenolic compounds such as coumarins, phenolic acids, flavonols and flavan-3-ols [16].

Bacterial infection simulation in the co-culture “race for the surface” assay revealed that addition of 100 μg/mL of either PSRE or PACN in the medium was effective in preserving viability of human gingival fibroblasts in the presence of both *S. aureus* and *A. actinomycetemcomitans* (Figure 2). Similarly, B-type linked proanthocyanidin-coated surfaces are shown to inhibit bacterial spreading and promote survival of mammalian cells [47]. The mechanism proposed to explain the activity is bacterial attachment and biofilm formation prevention by prodelphinidin-rich proanthocyanidins. In our experimental model, a similar efficiency was achieved by PSRE and PACN solutions, indicating that interaction of soluble compounds with the walls of bacteria also could mediate bacterial adhesion and mammalian cell protection. Accordingly, these results are very promising support to the use of natural extracts as an effective alternative antibacterial compound able to preserve the naïve tissue.

Investigation of gingival tissue protecting properties of PSRE and PACN in the bacterial LPS-mediated inflammation model revealed that both preparations efficiently prevent necrosis and apoptosis of fibroblast cells. Both substances were more efficient in decreasing the executing caspase-3 activity compared with the effect on apoptosis triggering caspase-8. However, PSRE was less efficient in suppressing caspase-8 activity than PACN, indicating that the latter had both upstream and downstream targets in the apoptotic cascade. The antiapoptotic activity of proanthocyanidins from grape seeds including decrease in executing caspases-3 and 9 was reported in a rotenone-induced neurotoxicity model of SH-SY5Y cells [48]. However, exposure of human colorectal carcinoma cells HCT-116 to proanthocyanidins from the same source significantly upregulated mRNAs encoding caspase-2, caspase-3 and caspase-9 [49]. Another study reports apoptosis induction in lung cancer cells NCI-H460 via stimulation of caspase-3 and mitochondrial cytochrome c release by gallic acid, one of the important constituents of PSRE [50]. Such controversial data suggest that the effect of PSRE and proanthocyanidins on apoptotic signaling pathways is cell type-dependent and they might have opposite effects in cancerous and non-cancerous cells as well as in different toxicity models.

Evaluation of pro-inflammatory cytokine secretion and gene expression revealed that PSRE and PACN suppress at least three different inflammatory processes: cytokine secretion (IL-8 from gingival fibroblasts and IL-6 from bone marrow-derived macrophages), inflammatory gene expression (IL-1β, iNOS and COX-2) and macrophage conversion to pro-inflammatory M1 phenotype related to the tissue loss in periodontitis. Downregulation of COX-2 coding mRNA in mononuclear leukocytes and PGE2 release from gingival fibroblasts indicate suppression of the prostaglandin inflammatory pathway. PGE2 is the most prominent in the pathogenesis of periodontitis among prostaglandins [51,52]. PGE2 is involved in the stimulation of inflammatory mediators and MMPs, as well as osteoclast formation via receptor activator of nuclear factor-κB ligand (RANKL) [52,53]. IL-6 and IL-1β also mediate bone resorption via osteoclasts activation [54], and increase in iNOS leads to reactive nitrogen species-mediated apoptosis of gingival fibroblasts [55]. By suppressing these inflammatory pathways, PSRE and PACN are expected to significantly improve condition and survival of periodontal tissues. Similar antiinflammatory activity of PSRE together with *Coptis chinensis* root extract was recently shown in LPS-stimulated RAW 264.7 cells [56]. The extract combination significantly decreased the levels of iNOS, PGE2, TNF-α, IL-1β and IL-6 in RAW 264.7 macrophages, and the results were also confirmed in vivo in a paw oedema rat model. Although the study reported lower levels of TNF-α secretion from LPS-stimulated RAW 264.7 cells, in our study, we did not observe significant changes on TNF-α gene expression in both LPS-stimulated leukocytes and LPS/IFN-γ-stimulated macrophages after PSRE and PACN treatment. Proinflammatory cytokine TNF-α plays a critical role not only in inflammatory cell migration, but also in both innate and adaptive immune responses, by up-regulating antigen presentation and the bactericidal activity of phagocytes [57,58]. In periodontitis, TNF-α is one of the key signals initiating several signaling pathways leading to chemotaxis of other inflammatory cells, tissue destruction and osteoclast formation [59,60]. The fact that PSRE and PACN had no effect on TNF-α expression level while suppressing several other related genes indicate the targets of the substances are located either downstream of the TNF-α signal or in the TNF-a excluding pathway.

Although antiinflammatory properties of PACN and PSRE revealed in the study were of comparative levels, PACN had stronger efficiency in suppressing caspases and preventing mediator release. Stronger anti-inflammatory activity of PACN might be due to greater amounts of prodelphinidins. These compounds possess higher antioxidant capacity and share certain important structural peculiarities, namely hydroxyl groups in B ring (especially in C4’ position and catechol group), hydroxyl groups in the A ring at the C5 and C7 positions [61].

## 5. Conclusions

In conclusion, both PSRE and PACN revealed antibacterial and antiinflammatory efficiency in periodontitis mimicking conditions. However, the combination of strong pathogen-selective antibacterial, antiinflammatory and gingival tissue protecting properties of PACN suggests this preparation as a potential candidate for treatment and prevention of periodontal disease.

## Figures and Tables

**Figure 1 nutrients-11-02829-f001:**
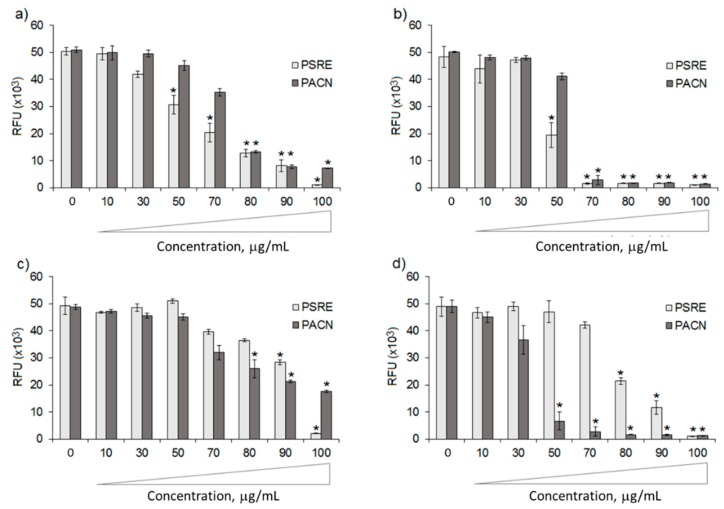
Antibacterial activity of *Pelargonium sidoides* root extract (PSRE) and proanthocyanidins from PSRE (PACN) towards *Staphylococcus aureus* (**a**), *S. epidermidis* (**b**), *Escherichia coli* (**c**) and *Aggregatibacter actinomycetemcomitans* (**d**). RFU—relative fluorescence units. Bars represent means and standard deviations of six experimental replicates. * = *p* < 0.05 vs. untreated control.

**Figure 2 nutrients-11-02829-f002:**
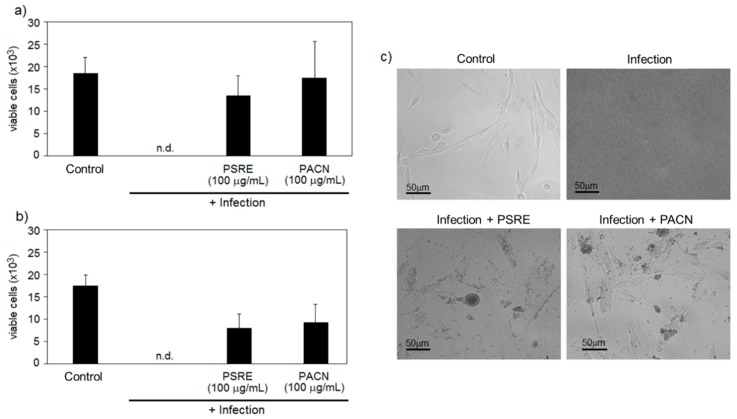
The effect of *Pelargonium sidoides* root extract (PSRE) and proanthocyanidins from PSRE (PACN) on human gingival fibroblast viability in co-culture “race for the surface” assay. Human gingival fibroblasts were infected with *A. actinomycetemcomitans* (**a**) or *S. aureus* (**b**); bars represent means and standard deviations of six experimental repeats. “Control” represents not infected cells, and “n.d.” means there were no detectable cells in the samples. In (**c**), there are representative images of human gingival fibroblasts with *A. actinomycetemcomitans* after 24 h of infection.

**Figure 3 nutrients-11-02829-f003:**
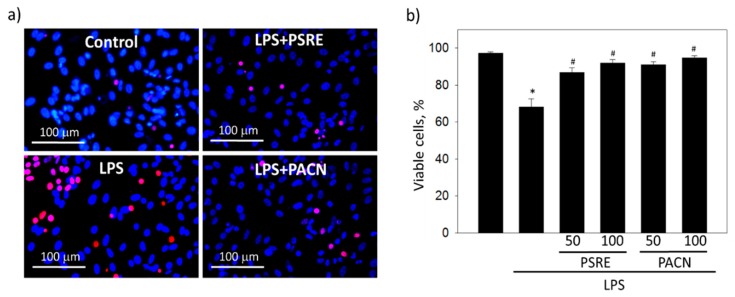
The effect of *Pelargonium sidoides* root extract (PSRE) and proanthocyanidins from PSRE (PACN) on the viability of gingival fibroblasts affected by bacterial lipopolysaccharide (LPS). (**a**) Representative fluorescent images after cell treatment with LPS and 100 μg/mL PSRE or PACN and nuclear viability staining; propidium iodide-positive necrotic nuclei are red, all nuclei are stained blue with Hoechst33342. (**b**) Quantitative viability results. Fifty and 100 represent the concentrations (μg/mL). The data are presented as means plus standard deviation of five experiments. *—significant difference compared to the untreated control, and #—significant difference compared to the LPS only treatment, *p* < 0.05.

**Figure 4 nutrients-11-02829-f004:**
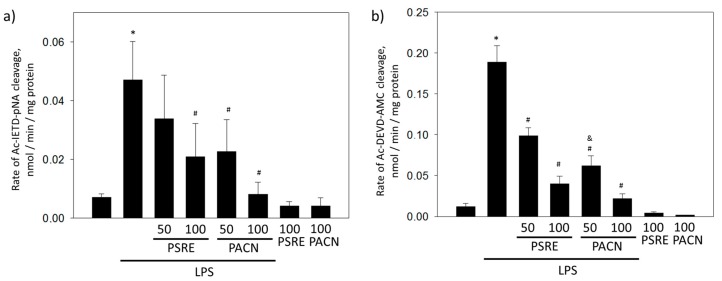
The effect of *Pelargonium sidoides* root extract (PSRE) and proanthocyanidins from PSRE (PACN) on LPS-induced caspase-8 (**a**) and caspase-3 (**b**) activation in gingival fibroblasts. Fifty and 100 represent the concentrations expressed in μg/mL. The data are presented as means with standard deviation of seven experimental repeats. *—significant difference compared to untreated control, #—significant difference compared to LPS only treatment and &—significant difference compared to LPS plus 50 μg/mL PSRE treatment, *p* < 0.05.

**Figure 5 nutrients-11-02829-f005:**
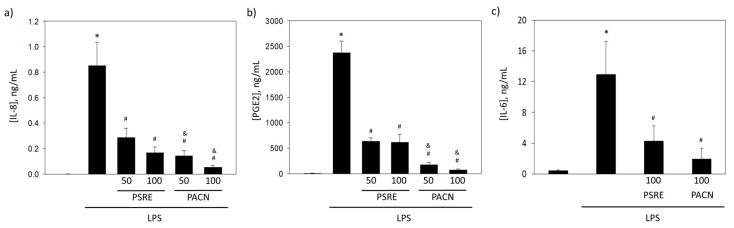
The effect of *Pelargonium sidoides* root extract (PSRE) and proanthocyanidins from PSRE (PACN) on (**a**) interleukin-8 (IL-8) and (**b**) prostaglandin E2 (PGE2) secretion from gingival fibroblasts, and (**c**) interleukin-6 (IL-6) secretion from peripheral blood mononuclear cells after LPS treatment. Fifty and 100 represent the concentrations (μg/mL). The data are presented as means and standard deviations of seven experiments. *—significant difference compared to untreated control, #—significant difference compared to LPS only treatment and &—significant difference compared to LPS plus a 50 μg/mL PSRE treatment, *p* < 0.05.

**Figure 6 nutrients-11-02829-f006:**
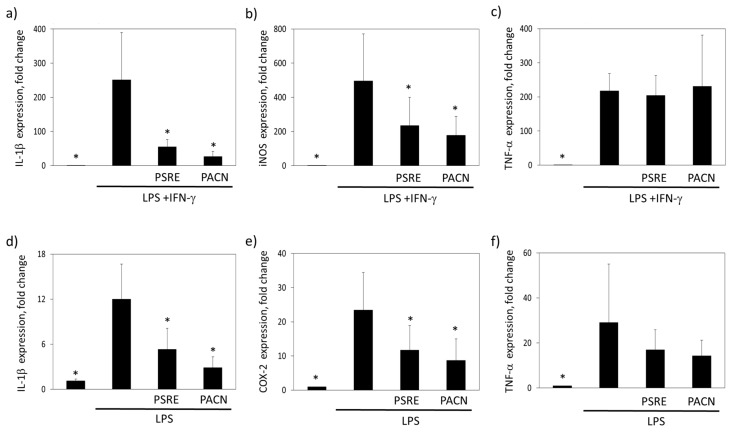
The effect of *Pelargonium sidoides* root extract (PSRE) and proanthocyanidins from PSRE (PACN) on proinflammatory gene expression in bone marrow-derived macrophages (**a**–**c**) and peripheral blood mononuclear cells (**d**–**f**) after LPS or LPS and IFN- stimulation. (**a**,**d**) IL-1β (**a**,**b**) iNOS, (**c**,**f**) TNF- and (**e**) COX-2. The data are expressed as a fold change of glucose-6-phosphate isomerase gene transcription and presented as mean ± SD of three independent measurements. *—significantly different from the LPS-treated samples (ANOVA followed by a Tukey’s multiple comparison test, *p* < 0.05).

**Figure 7 nutrients-11-02829-f007:**
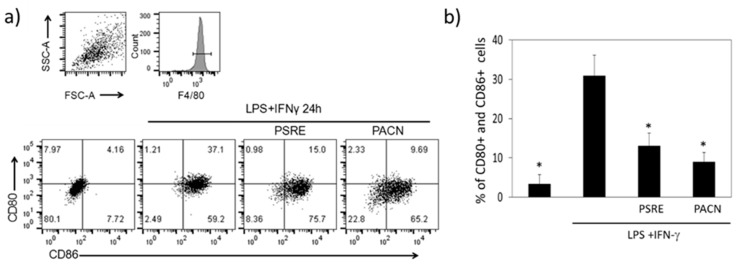
Expression of proinflammatory cell surface markers CD80 and CD86 analyzed by flow cytometry 24 h after treating LPS + IFNγ-activated BMDMs with PSRE and PACN. (**a**) Characteristic mouse macrophage marker F4/80-positive cells were gated for double CD80 and CD86 analysis as a measure of M1 macrophage phenotype (top right small quadrant). Representative plots of a total of three independent experiments in three replicates are presented in the bottom. (**b**) Mean ± SD of three independent measurements in three parallels. Differences between the measurements were tested using a one-way ANOVA followed by a Tukey’s multiple comparison test. *—significantly different from the LPS and IFN-γ treatment (*p* < 0.05).

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
