# Peer review of "Investigation of Antibacterial and Antiinflammatory Activities of Proanthocyanidins from Pelargonium sidoides DC Root Extract"

_nutrients, 2019, doi:10.3390/nu11112829_

Round 1

Reviewer 1 Report

The manuscript describes study aimed on investigation of antibacterial and anti-inflammatory activities of Pelargonium sidoides DC root extract. While the experiments seem to be well performed there are some significant issues in study design and interpretation of obtained results.

The major issue in study design is the use of cells (both cell lines and primary cells) originating from different species. First, author used human gingival fibroblasts, then switched to rat gingival fibroblasts, and finished with human PBMCs and BMDMs from mouse. What is the rationale for using cells from three different species? I understand that some processes might be similar in all three types of cells, but, in my opinion, at the stage of study design the use of only one species should have been considered. There is no data or even information regarding influence of PSRE/PACN on all cells used in the study. Supplementary data include survival curves for rat fibroblasts only, and the same data would be necessary for other types of cells, provided that use of cells from different species was justified (see pt. 1) It is not clear what type of controls were used in the experiments. Does "control" mean cells untreated with anything, just cultured in plain medium or this refers to cells treated only with PSRE/PACN at indicated concentration? If so then which substance was used in a particular experiment, PSRE or PACN? p.9 l.351-352 Authors claim that "none of the substances was toxic to gingival fibroblasts when applied for 24h". The supplementary figure S1 does not support this claim, especially in the case of PSRE, where estimated 10% of cells die after 24h treatment. Authors draw very far reaching and not correct conclusion regarding the role of LPS in sensitivity of bacteria to tested substances. First, authors summarize that PSRE/PACN indicate stronger toxicity to pathogens than to non-pathogenic E. coli (p.7 l.266-267). In the discussion section authors suggest that LPS-affinity might be one of the possible explanations of specific targeting of PACN to pathogenic strains (p.13 l.502-503). First of all, the conclusion about stronger toxicity to pathogens can be only limited to bacteria used in the study and cannot be extended as general phenomenon. Second, if the affinity of PACN to LPS was responsible for killing of bacteria why E. coli is more resistant than other bacteria used in the study. Third, LPS is a compound of cell walls of Gram-negative bacteria (here E. coli and A. actinomycetemcomitans), not Gram-positive (here S. aureus and S. epidermidis). So the conclusion that LPS is responsible for enhanced killing of pathogens is wrong. The description of statistical analysis is not precise. Authors provide information that 3-7 replicates were used. Report exact number of repeats of particular experiments.

Author Response

Authors Reply R1

Reviewer #1

The manuscript describes study aimed on investigation of antibacterial and anti-inflammatory activities of Pelargonium sidoides DC root extract. While the experiments seem to be well performed there are some significant issues in study design and interpretation of obtained results.

Authors are very thankful for the time and effort reviewing our study. Here we provide the point-by-point reply to the comments in italics.

The major issue in study design is the use of cells (both cell lines and primary cells) originating from different species. First, author used human gingival fibroblasts, then switched to rat gingival fibroblasts, and finished with human PBMCs and BMDMs from mouse. What is the rationale for using cells from three different species? I understand that some processes might be similar in all three types of cells, but, in my opinion, at the stage of study design the use of only one species should have been considered.

We completely agree with the point that to perform all the experiments on the cells from the same species would make the study design more rational. However, the current manuscript is the outcome of collaboration among three research groups while performing an international project, and combines the assays and research objects that are used routinely in each of the participating laboratories.  In support of the current design we could add that although the data produced on rodent gingival fibroblast cell cultures are not human cell data, however, they are generated by using primary cells, not cell lines that have genetic differences compared with the original primary tissue cells. Therefore, we still find the data provide useful knowledge about the biological activity of Pelargonium sidoides root extract and isolated proanthocyanidins.

There is no data or even information regarding influence of PSRE/PACN on all cells used in the study. Supplementary data include survival curves for rat fibroblasts only, and the same data would be necessary for other types of cells, provided that use of cells from different species was justified (see pt. 1)

We have performed the dose-dependent toxicity evaluation of PSRE and PACN on human peripheral blood mononuclear cells and bone-marrow derived macrophages. The data are now presented in Supplementary Figure S2 (A) and Supplementary Figure S3 (A).

It is not clear what type of controls were used in the experiments. Does "control" mean cells untreated with anything, just cultured in plain medium or this refers to cells treated only with PSRE/PACN at indicated concentration? If so then which substance was used in a particular experiment, PSRE or PACN? p.9 l.351-352

Regarding the term “control”, it means cells were cultured under the same conditions without any treatment. In figures, the control samples are presented either with the label “Control”, or without any label.

Authors claim that "none of the substances was toxic to gingival fibroblasts when applied for 24h". The supplementary figure S1 does not support this claim, especially in the case of PSRE, where estimated 10% of cells die after 24h treatment.

As indicated in the Supplementary Figure S1, 50 and 100 mg/mL PSRE and PACN did not cause statistically significant changes in viability of gingival fibroblasts compared to the untreated control. The change in necrotic cell values in samples affected by this concentrated PSRE or PACN were within the zone of standard deviation, thus, we have considered the conditions as not toxic.

Authors draw very far reaching and not correct conclusion regarding the role of LPS in sensitivity of bacteria to tested substances. First, authors summarize that PSRE/PACN indicate stronger toxicity to pathogens than to non-pathogenic E. coli (p.7 l.266-267). In the discussion section authors suggest that LPS-affinity might be one of the possible explanations of specific targeting of PACN to pathogenic strains (p.13 l.502-503). First of all, the conclusion about stronger toxicity to pathogens can be only limited to bacteria used in the study and cannot be extended as general phenomenon.

We agree that the conclusions might only include the bacteria tested in the study. However, our previous study demonstrated higher sensitivity of pathogen P. gingivalis to PACN compared to commensal S. salivarius, and other authors have discovered similar selective action of proanthocyanidins derived from berries as reviewed by Lacombe and Wu. The discussion is now extended and includes more findings about this issue.

Second, if the affinity of PACN to LPS was responsible for killing of bacteria why E. coli is more resistant than other bacteria used in the study. Third, LPS is a compound of cell walls of Gram-negative bacteria (here E. coli and A. actinomycetemcomitans), not Gram-positive (here S. aureus and S. epidermidis). So the conclusion that LPS is responsible for enhanced killing of pathogens is wrong.

Thanks for the remark. By checking the text after Reviewer’s comment, we noticed that the conclusions related to antibacterial activity and LPS are contradictory and we have modified the text as suggested. We have also removed the sentence related to the LPS role in enhancing extract activity as we agree with Reviewer’s comment that this hypothesis is not sustainable with our results. We have extended the discussion including some other mechanisms described in the literature to justify the differences between different response of pathogens and non-pathogen strains to PSRE and PACN.

The description of statistical analysis is not precise. Authors provide information that 3-7 replicates were used. Report exact number of repeats of particular experiments. 

The exact number of repeats is now provided in each Figure legend.

Reviewer 2 Report

This topic is very interesting. This manuscript included in vitro study, in the future, in vivo experimental periodontitis, it needed to confirm these results.

Author Response

Reviewer #2:

This topic is very interesting. This manuscript included in vitro study, in the future, in vivo experimental periodontitis, it needed to confirm these results.

Authors Reply R2

Authors are very thankful for the time and effort reviewing our study, for the positive evaluation and future research suggestion.

Reviewer 3 Report

Dear authors,
I congratulate you on the originality and scientific rigor of your manuscript.
I only advise you to double check the graphic release of some points because it is not homogeneous and to add a paragraph at the beginning of the discussion. It would be interesting, in fact, to treat the use of other bacteria as a therapy against periodontopathogens (especially in light of current antibiotic resistance). I recommend you to cite this work on the topic:

Patini R et al. Evaluation of predation capability of periodontopathogens bacteria by Bdellovibrio Bacteriovorus HD100. An in vitro study. Materials 2019, 12, 2008.

Best regards

Author Response

Reviewer #3

I congratulate you on the originality and scientific rigor of your manuscript.

I only advise you to double check the graphic release of some points because it is not homogeneous and to add a paragraph at the beginning of the discussion. It would be interesting, in fact, to treat the use of other bacteria as a therapy against periodontopathogens (especially in light of current antibiotic resistance). I recommend you to cite this work on the topic:

Patini R et al. Evaluation of predation capability of periodontopathogens bacteria by Bdellovibrio Bacteriovorus HD100. An in vitro study. Materials 2019, 12, 2008.

Authors Reply R3

Authors are very thankful for the time and effort reviewing our work and for the positive attitude. We have mentioned the suggested bacterial strategy to defeat the pathogenic strains, which is very interesting indeed. However, we have decided not to discuss it in more detail because is not directly relevant to the results of our study.

Round 2

Reviewer 1 Report

All my questions have been answered and the manuscript accordingly corrected.